# Analysis of Spatial Decorrelation of Small-Scale Tropospheric Delay Using High-Resolution NWP Data [note 1]

**DOI:** 10.3390/s23031237

**Published:** 2023-01-21

**Authors:** Jan Erik Håkegård, Nadezda Sokolova, Aiden Morrison

**Affiliations:** SINTEF Digital, Strindveien 4, 7032 Trondheim, Norway

**Keywords:** non-nominal troposphere, troposphere gradient, NWM, GNSS

## Abstract

This paper contains results from a study where numerical weather product (NWP) data provided by MET Norway were used to estimate the differential zenith tropospheric delay (dZTD) for an area including Scandinavia, Finland and the Baltic countries. The NWP data have a high spatial resolution of 2.5×2.5 km, and the estimated dZTD for the grid positions allows for calculation of the tropospheric gradient on short baselines. The results give an indication of how large dZTD values for baselines of up to 20 km can be, and of where the largest events are located within the coverage area. One year of data were processed, and dZTD values up to 18 cm with baselines were detected. Preliminary results comparing the NWP-based results with GNSS-based results are included. The motivation for this investigation was to better understand the characteristics of this phenomenon as a preamble to a later investigation of how it might impact GNSS-based navigation systems with integrity support in these regions.

## 1. Introduction

The troposphere constitutes an important error source for GNSS systems, as refraction through the troposphere delays the transmission of satellite signals. While the tropospheric delay phenomenon is well understood, accurate modeling of its spatial variation during anomalous tropospheric events and the impact of any unmodelled residuals remains a challenge. Under nominal conditions, the majority of the tropospheric delay can be eliminated by a tropospheric model such as [1,2] and/or through differential processing. However, during heavy rainfall or other severe weather conditions, tropospheric anomalies may occur, causing high spatial variation [3,4] of the tropospheric delay that cannot be easily mitigated, resulting in residual errors that can affect the navigation system’s accuracy and integrity. This is of particular concern for both the existing GNSS augmentation techniques, such as ground-based augmentation systems (GBAS) [5,6] that rely on correction generation based on local observations and modeling the effect on the user side, and the evolving high precision GNSS augmentation systems, such as the network real time kinematic (NRTK) implementations [7,8] aiming at integrity support to cover the needs of new user groups, e.g., in future autonomous transportation operations where the observed residuals are interpolated to the position of the user or a virtual reference station (VRS).

In some techniques, such as PPP and PPP-RTK, the tropospheric error is estimated on the user’s side. This is not the approach taken by NRTK and similar methods, where the atmospheric biases are interpolated to the user’s location. In the case of PPP/PPP-RTK, the vertical tropospheric gradient is the primary concern; however, in high dynamic user scenarios, the horizontal variation is also likely of interest. In this paper, we focus on characterization of the horizontal dZTD with regard to NRTK and similar system operations, particularly if and when integrity support is considered.

The effects of the troposphere on GNSS signals can be exploited to obtain information about the state of the troposphere, and in particular, the distribution of water vapor. This is currently a well-established field of research and operation referred to as GNSS meteorology [9]. While GNSS continuously operating reference station (CORS) networks represent a very important and accurate data source for observing the zenith total delay (ZTD) [10] and its hydrostatic and wet components (ZHD and ZWD, respectively), the density of such networks does not allow for studying the spatial decorrelation of the tropospheric delay on a smaller scale. As in many techniques and applications, the distance between the user and the ground network/VRS is much less than ten kilometers, there is a need to study the spatial decorrelation of the tropospheric delay during anomalous conditions on shorter baselines. This was carried out in this work by using high spatial resolution numerical weather model (NWM) data. NWMs use a large variety of meteorological observations to describe atmospheric dynamics and generate weather forecasts. One of the advantages of the NWMs is the capability of retrieving ZTD and horizontal gradients with high spatial resolution covering a large geographical area.

NWM data/numerical weather products (NWPs) are issued by meteorological organizations such as the ECMWF (European Centre for Medium-Range Weather Forecasts) for Europe, and the Norwegian Meteorological Institute (MET Norway) for Norway and the surrounding areas. In this study, we estimated the differential ZTD (dZTD) from NWP data provided by MET Norway, the resolution of which is 2.5 × 2.5 km. The dZTD is calculated as the difference between ZTD values for baselines of up to 20 km.

It is noted that other open data sources exist for zenith tropospheric delay and gradients that are based on NWP data. The first of these is the Vienna Mapping Function (VMF), for which the open-access data [11] contain ZHD and ZWD data and north–south and west–east gradients globally with a grid resolution of $1^{\circ }\times 1^{\circ }$. With this grid configuration, the resolution in latitude is approximately 111 km and in longitude is between 30 km and 70 km when considering latitudes in the range $50^{\circ }$–$70^{\circ }$N. The resolution of the VMF data is unfortunately too coarse to be used to estimate differential zenith delays with baselines lower than 20 km, as was done in our study.

In the following sections, the NWP data are briefly described and the model used to estimate tropospheric delays from NWP data is reviewed. After this information is introduced, analysis results from one year of NWP data are presented. Then five days of GNSS measurements from two stations are used to do a preliminary assessment of the accuracy of the NWP-based estimates, followed by discussion of outcomes from the analysis. The final section of the article covers conclusions of the work presented.

## 2. Model Description

### 2.1. Data Sources

MET Norway disseminates three distinct NWPs (https://thredds.met.no/thredds/metno.html, accessed on 24 November 2022). The MetCoOp Ensemble Prediction System (MEPS) provides 30 ensemble members every six hours with lead times of up to 66 h for a spatial grid with 2.5 km resolution and 65 vertical layers. In addition to publishing data files with 30 ensemble members, which is useful for assessing probabilities, it also provides files containing the ensemble member that on average best coincide with observed weather. These are the files that are considered in the analysis presented in this paper. Arome Arctic provides one single run with up to 66 h lead time also for a spatial grid of 2.5 km resolution and 65 vertical layers. Finally, post-processed products, including only surface parameters, are provided. Figure 1 shows the coverage areas of the three products. The green line shows the MEPS coverage area, the blue line the Arome Arctic coverage area, and the red line the post-processed products coverage area.

To estimate the tropospheric delay, a 3D model was used. Hence, only the MEPS data and the Arome Arctic data were considered. Although the two data sets contain different parameters, they both include the information required to estimate the tropospheric delay. Although the coverage areas are overlapping and the spatial resolution is the same, the grid points are placed at different locations. In order to evaluate the congruity of the two data sets, an initial comparison of the calculated ZTD values considering the area shown in yellow in Figure 2 was carried out. The grid points of the MEPS data set were mapped on to the grid points of the Arome Artic data set using linear scattered interpolation, and the difference between the two was calculated. Figure 3 shows the probability density function (PDF) of the difference. The curve indicates that the difference is within 2 cm for 0.11% of the grid points. The PDF was calculated based on 100 data files, or about 10 million sample points. The curve confirms that using MEPS and AROME Arctic will provide comparable results when estimating tropospheric delay, and that it is the coverage area of interest that is the decision criterion for which data set to use. In this study, the MEPS data set was selected, as it covers the entire Norwegian mainland and surrounding areas.

### 2.2. ZTD Calculation Based on the NWP Data

In this section, the equations used to derive the tropospheric delay based on NWP data are reviewed. The parameters used as input are listed in Table 1. The temporal resolution is 3 h. Only the forecast with the shortest lead time is considered in this study, as it is assumed to be the most accurate one. The horizontal grid of longitude and latitude positions is 739×949 points.

The ZTD is calculated by integrating along the vertical model: (1)$$ZTD=10^{-6}\sum_{k=1}^{L}d\left(k\right)N\left(k\right)$$
where d(k) is the thickness of vertical layer *k*, N(k) is the refractivity of level *k*, and *L* is the number of vertical layers. Hence, to estimate the ZTD, the thicknesses of the layers and the refractivity of the layers must be estimated.

#### 2.2.1. Estimation of the Thicknesses of the Vertical Layers

The height of each of the vertical model’s layers h(k) is estimated from the air pressure and temperature of said layer. The procedure is provided by MET Norway and consists of integrating from the ground and up [12]: (2)$$h\left(k\right)=h\left(k+1\right)+\frac{R\cdot T(k)}{g}\ln\left(\frac{p(k+1)}{p(k)}\right)$$
where R=287.058 J/kgK, *T* is the temperature in Kelvin found in the data, and *p* is the pressure in Pascal. The last level h(L) is the surface height $h_{s}$ found using the following equation: (3)$$h_{s}=\frac{\theta_{0}}{g}$$
where *g* is the standard gravity. The surface geopotential $\theta_{0}$ is provided in the data. The pressure at each model level is not included in the data but can be estimated using the hybrid model’s parameters. The air pressure at model level *k* is given by: (4)p(k)=ap(k)+b(k)·ps

The components of the hybrid model ap(k) and b(k) and the surface air pressure ps are all available in the data. When the height of each level is derived, we also have the thickness of each level.

#### 2.2.2. Estimation of the Refractivity Index

The refractivity index is defined as [13]: (5)$$N=N_{dry}+N_{wet}=77.6\frac{p_{d}}{T}+72\frac{e}{T}+3.75\times 10^{5}\frac{e}{T^{2}}$$
where $p_{d}$ is the dry air pressure in hPa and *e* is the water-vapor pressure in hPa. In this subsection, we omit the model’s layer parameter *k* for simplicity. The total atmospheric pressure is the sum of the dry and water-vapor pressure:(6)$$p=p_{d}+e$$

The temperature is available in the data, and the pressure can be estimated using Equation (Equation 4). The water-vapor pressure can be found from the relative humidity RH and the saturation vapor pressure $e_{s}$ [14]:(7)$$e=\frac{1}{100}RH\cdot e_{s}$$

Several different formulas for estimating the saturation vapor pressure are available in the literature. Here, the widely used Magnus formula is employed [14]:(8)$$e_{s}\approx 611e^{\left(\frac{17.67(T-T_{0})}{T-29.65}\right)}$$
where $e_{s}$ is in Pascal. $T_{0}$ is the reference temperature (typically 273.15 K).

The relative humidity is defined as the ratio of the actual water-vapor pressure *e* to saturation vapor pressure $e_{s}$ or as the ratio of the actual water vapor dry mass mixing ratio *w* to the saturation mixing ratio $w_{s}$ at the ambient temperature and pressure. The two definitions are related as follows:(9)$$w=\frac{R_{d}}{R_{v}}\frac{e}{p-e},w_{s}=\frac{R_{d}}{R_{v}}\frac{e_{s}}{p-e_{s}}$$
where the specific gas constants for dry air $R_{d}=287.058$ J/(kgK) and vapor $R_{v}=461.5$ J/(kgK). The two definitions are essentially equivalent in the cases where $e<e_{s}<<p$.

The specific humidity can be defined as the mass mixing ratio of water vapor in air, defined as:(10)$$q=\frac{m_{v}}{m_{d}+m_{v}}=\frac{w}{w+1}\approx w$$

The relative humidity can then finally be estimated as: (11)$$RH=100\frac{w}{w_{s}}=100\frac{R_{v}}{R_{d}}\frac{pq}{e_{s}}$$
and the saturation vapor pressure as:(12)$$e=\frac{R_{v}}{R_{d}}pq=1.6077pq$$

The refractivity index then becomes:(13)$$N=N_{dry}+N_{wet}=77.6\frac{p_{d}}{T}+115.75\frac{pq}{T}+6.0288\times 10^{5}\frac{pq}{T^{2}}$$

Care must be taken with the units, as the pressure *p* using the MET data is in Pa, whereas Equation (Equation 5) takes pressure parameters $p_{d}$ and *e* in hPa. Hence, $p_{d}$ and *p* in Equation (Equation 13) are in hPa.

The zenith hydrostatic and wet delays can then be estimated as:(14)$$ZHD=77.6\times 10^{-6}\sum_{k=1}^{L}d\left(k\right)\frac{p_{d}\left(k\right)}{T(k)}$$
(15)$$ZWD=10^{-6}\sum_{k=1}^{L}d\left(k\right)\left(\frac{115.75}{T(k)}+\frac{6.0288\times 10^{5}}{T^{2}\left(k\right)}\right)p\left(k\right)q\left(k\right)$$

### 2.3. Differential ZTD

The differential ZTD (dZTD) is calculated as the difference between the estimated ZTD values. dZTD values for baselines up to 20 km are considered to be of interest here. It would be too computationally expensive to calculate all of the dZTD values for baselines of up to 20 km. Therefore, a grid with resolution 20 × 20 km is generated, and the maximum and minimum ZTD values within each 20 × 20 km pixel are found and stored together with the corresponding baseline. The coverage area after this constraint consists of 69×87 pixels.

Only horizontal dZTD is considered. Due to the varying surface height, dZTD values are calculated for seven heights above sea level: 10, 200, 400, 600, 800, 1000, and 1600 m. Only the results from the height providing the highest dZTD value for each pixel are retained.

### 2.4. Relationship between Differential ZTD and Tropospheric Gradient

The slant tropospheric delay (STD) of GNSS signals can be expressed as:(16)$$\begin{array}{c} STD\left(el,az\right)=m_{h}\left(el\right)ZHD+m_{w}\left(el\right)ZWD\\ +m_{g}\left(\cos(az)N+\sin(az)E\right), \end{array}$$
where az and el are the azimuth and elevation angles; *N* and *E* are the north and east gradient components; and $m_{h}$, $m_{w}$, and $m_{g}$ are mapping functions. The gradient represents the first-order asymmetry of the tropospheric delay. Several mapping functions exist. Most commonly used are CfA-2.2, Ifadis, mapping temperature test (MTT), Neill’s mapping function (NMF), Global mapping function (GMF), and Vienna mapping function (VMF1) [15].

The gradient components can be split into hydrostatic and wet components in the same way as the ZTD. For the wet components, the following is approximately true [16]:(17)$$\begin{array}{c} E_{w}\approx C\frac{\partial ZWD}{\partial x}=\frac{C}{R}\frac{\partial ZWD}{\partial \lambda }\frac{1}{\cos\phi }\\ N_{w}\approx C\frac{\partial ZWD}{\partial y}=\frac{C}{R}\frac{\partial ZWD}{\partial \phi }, \end{array}$$
where λ and ϕ denote the station latitude and longitude, and *R* is the Earth’s radius. *C* is related to the scale height of the wet refractivity gradient and is set to 4 km in [16].

In this study, we used a slightly different approach. The gradient *G* was approximated as the difference between ZWD values divided by the baseline:(18)$$G\approx \frac{ZWD\left(x_{1},y_{1}\right)-ZWD\left(x_{2},y_{2}\right)}{d(\left(x_{1},y_{1}\right),\left(x_{2},y_{2}\right))},$$
where $ZWD(x_{1},y_{1})$ and $ZWD(x_{2},y_{2})$ are the maximum and minimum ZWD estimates within the 20×20 km pixel, and $d(\left(x_{1},y_{1}\right),\left(x_{2},y_{2}\right))$ is the distance. As the goal was not to estimate the STD using (Equation 16), we did not split the gradient into north and east components. A scale height *C* was not set for the same reason.

## 3. Results

### 3.1. MEPS Coverage Area

dZTD values were calculated based on files from MET Norway issued every 3 h during 2021, totaling 2920 epochs. Figure 4 shows the maximum estimated dZTD in each epoch for the MEPS coverage area shown in Figure 1. Hence, the curve does not correspond to one location, but to the maximum estimated dZTD value for the entire coverage area for each epoch. The curve clearly shows that the highest dZTD values were observed during the summer months, from mid May to the end of August.

Table 2 shows the five highest estimated dZTD values during 2021, including the times and the corresponding baselines. The largest estimated dZTD event occurred 14 July, just north of Gothenburg, and the estimated dZTD was 18.1 cm with a baseline length of 17.7 km, corresponding to a gradient of 10.2 mm/km using (Equation 18).

Figure 5 shows the horizontal gradients in mm/km for the largest dZTD events, again using (Equation 18). The steepest gradient of 22.8 mm/km was estimated on 16 July, with a dZTD of 11.4 cm and a baseline of 5 km. It should be noted that these gradients are calculated for the maximum dZTD value per pixel. It is likely that there exist larger gradients with shorter baselines.

Figure 6 shows where the 50 strongest dZTD events occurred. Almost all of them happened in the eastern part of the coverage area, i.e., in Finland and the Baltic countries. Only five were located in Sweden, one in Norway, and one in Skagerrak.

It was surprising that almost all events were in the eastern parts of the coverage area, and it was therefore investigated further. Two hypotheses can be made for the cause of this:

**Hypothesis** **1.**
*There are edge/border problems related to the data from MET Norway.*


**Hypothesis** **2.**
*There are meteorological explanations to the events.*


The methodology applied to test these hypotheses was as follows:Identify the four largest events in the data, of which three occurred in the south-eastern part of the coverage area (see Figure 6).Investigate the ZTD data and the meteorological parameters for these events.Check if the events occur at the border of the coverage area (edge problem).Check if the meteorological parameters support the dZTD estimates (meteorological explanation).

Figure 7 shows the ZTD values for the four events. The area covered by these figures is 60×60 km; the areas corresponding to the maximum and minimum ZTD values are in the middle. In the figures, there are no signs of any edge effect. It therefore seems that hypothesis 1 is wrong. Another observation from Figure 7 is that the extreme values are very local. In these areas, extreme rain events are often local in nature and short in duration. It is therefore not surprising that when the data samples providing the highest dZTD values are extracted from one year of data, the events are local in nature.

Next, the meteorological parameters used to estimate the delays are considered. Figure 8 illustrates how the three parameters *q*, *p*, and *T* vary for three of the events. The blue line corresponds to the location with maximum delay and the red line to the location with minimum delay within the 20×20 km pixel. As already shown, the ZWD is proportional to *p* and *q*, and inversely proportional to $T^{2}$. The curves clearly show how the humidity *q* varies along the vertical column for all four events. The lower right part of Figure 8 contains curves for an arbitrary pixel, where no event was detected. In this case, the variations in delay over the pixel are much lower, and the variation in humidity is much lower. Hence, it can be concluded that the meteorological parameters confirm the observation that most of the events during this year occurred in the eastern part of the coverage area.

The data shown in this section only include the maximum dZTD values over the entire coverage area at each epoch. It may therefore be the case that there are events with higher dZTD values in Norway that were not included because they were masked by even higher dZTD values at the same time outside Norway. To check this, we consider the data only for the mainland of Norway in the next section.

### 3.2. Analysis of Data from the Norwegian Mainland

Figure 9 shows the maximum estimated dZTD values for Norway. The highest value occurred on 14 July at noon, close to Halden. This is the same weather phenomenon that caused the 18.1 cm event at the same time a few kilometers further to the south in Sweden.

Table 3 lists the five largest dZTD events observed in Norway. The three largest ones were over 10 cm. Figure 10 illustrates the geographical distribution of the estimated dZTD values. As can be seen, they are scattered from the southern parts of Norway to Finmark in the north. Events 3 and 5 were in fact part of the same meteorological event, as they are from consecutive epochs and almost co-located. Events 2 and 4 are from consecutive epochs, but are located within different parts of Norway.

## 4. Comparison with GNSS Derived Measurements

In this section, we compare the ZWD and dZWD values estimated from NWP data with the ones derived from the GNSS measurements. Two GNSS receivers located close to Tromsø in Northern Norway were used (see Figure 11). The distance between the stations is 23.1 km, well above the resolution of the NWP grid. The GNSS data were processed using the GipsyX software.

This section contains preliminary results from this study. Five days of data were considered. The time period was selected as this was a period where the NWP data showed significant tropospheric activity. The temporal resolution of the GNSS derived data is 5 min, and one sample from each NWP file was used, giving a temporal resolution of 3 h. The NWP grid points closest to the GNSS receiver/station positions were used in the comparison.

Figure 12 shows the estimated ZWD values based on the GNSS-derived measurements and based on the NWP data for the two stations. The curves show good correlation between the two data sets, although not perfect. The mean and the standard deviation of the difference between the two are listed in Table 4.

In Figure 13, the differential ZWD between the two stations is illustrated. The estimated dZWD based on GNSS measurements shows larger variations than the dZWD estimates based on the NWP data. The maximum dZWD value derived from the GNSS measurements is about 2.15 cm, while it is about 0.44 cm based on NWP data.

## 5. Discussion

The high resolution of the MEPS data provides an opportunity to estimate local variations in the tropospheric delay based on NWP data that other sources with $1^{\circ }\times 1^{\circ }$ resolution do not support. The dynamic and in-homogeneous nature of the humidity of the troposphere requires data samples with high density for the linear expansion of the derivative at a certain point to be reasonably accurate. The high resolution is, however, a challenge computationally. The methodology used to estimate maximum dZTD values in this study gives an indication of how much the ZTD, and hence ZWD, may vary over distances of up to 20 km. It does, however, not involve an exhaustive search, as that would be too time consuming. Dividing the coverage area into 20 km pixels and only considering the maximum and minimum values within a pixel opens the possibility that two locations will exist within 20 km distance, but in different pixels, may provide higher dZTD values. Still, considering the large number of epochs and the large coverage area, it is considered unlikely that considerably higher dZTD values are missed.

Another feature of the method used in this study is that the maximum dZTD value within each pixel is found. This may not coincide with the highest gradient within the pixel, as lower dZTD values with shorter baselines may lead to steeper gradients. The reason for this choice was also to reduce the computational load. A continuation of the work could include selection of epochs and locations where large dZTD values are observed, and find the steepest gradients in those data sets.

Studies from other parts of the world reported spike events of up to 300 mm/km [5]. One reason for this is that convective rain events with higher intensity occur further to the south. Such events tend to be very local and of short duration. Moreover, the NWP data will represent an averaging over both the 2.5×2.5 km spatial area and over the 3 h time resolution. Measurements may therefore capture very local and short-duration events that the NWP data miss. This is illustrated by the comparison between the data based on GNSS measurements and on NWP data included in the previous section. The same effect can be observed when comparing NWP data with weather radar data: the radar data show higher maximum values for rain rates. The NWP data sets contain variability parameters that may be used to statistically interpolate in both the spatial and temporal domains, but these were not used in this study.

The results obtained from this study can be compared to ZWD estimates obtained from a cluster of GNSS receivers [17]. As the NWP data provide predictions of the meteorological parameters and do not constitute ground truth, there are bound to be differences in the results. Combining GNSS derived measurements and results from NWP data may, however, provide an improved estimation of the tropospheric gradients.

## 6. Conclusions and Future Work

In this study, a set of NWP data covering the year 2021 was used to find the maximum differential ZTD values for baselines of up to 20 km. First, two available NWP data sets (MEPS and AROME Arctic) were evaluated for consistency over their region of overlap. Within this overlapping area, it was found that the level of disagreement was almost universally below 2 cm in magnitude, indicating that either model could be used for subsequent analysis with this level of uncertainty, leading us to select the model with better coverage of our area of interest. The selected MEPS data set covers the Nordic countries and the Baltic region. The results show that dZTD values approaching 20 cm can be observed, and gradients of approximately 25 mm/km can be found. Most of the observed events occurred in the eastern parts of the coverage area within the considered one-year time period. The NWP data provide a higher spatial resolution of the tropospheric gradient than using estimates from the GNSS CORS networks alone, though a denser GNSS station network could in turn provide locally better estimates in that region. The observed dZTD and gradient values are limited compared to gradients that can be encountered in other regions; however, the implications for GNSS navigation users implementing navigation integrity remain.

While the NWP product data allow characterization of the spatial variation in short baselines, it is necessary to compare these high-resolution model-data-based results with GNSS derived measurements over a co-located dense network to characterize the residuals due to features not being captured by the model. In this paper, sample ZWD and dZWD results based on the evaluation of five days of the highest observed tropospheric activity using a pair of receivers have been presented. Within this preliminary evaluation, the GNSS measurements captured larger gradients than the NWP data. Further work is required to quantify this effect more accurately. It is therefore planned to extract additional periods of elevated tropospheric activity through processing and analyzing multiple years of data and to extend the comparative GNSS analysis to employ a dense cluster or clusters of receivers within the NWP model’s area of coverage. Based on the outcome of this further investigation, the implications of the observed tropospheric spatial gradient values in the context of GNSS systems/services with integrity support will be determined.

## Figures and Tables

**Figure 1 sensors-23-01237-f001:**
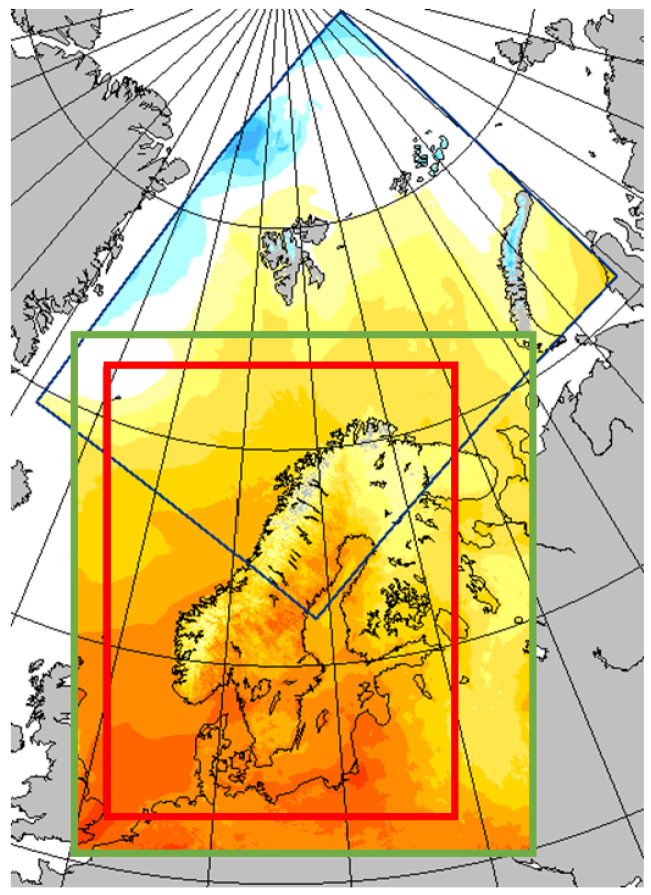
Coverage area for the NWP products. Blue square: Arome Arctic, green square: MEPS, red square: post-processed products (source: MET Norway).

**Figure 2 sensors-23-01237-f002:**
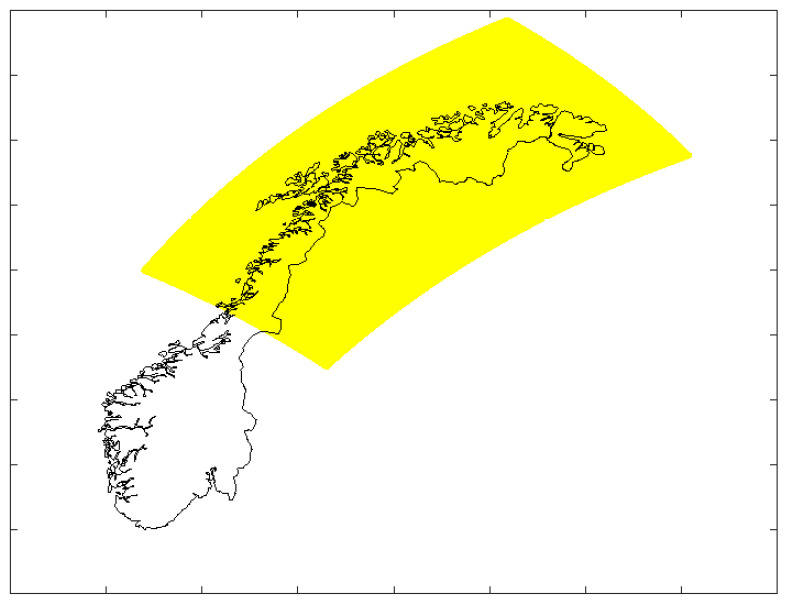
Coverage area used for comparing MEPS and Arome Arctic data.

**Figure 3 sensors-23-01237-f003:**
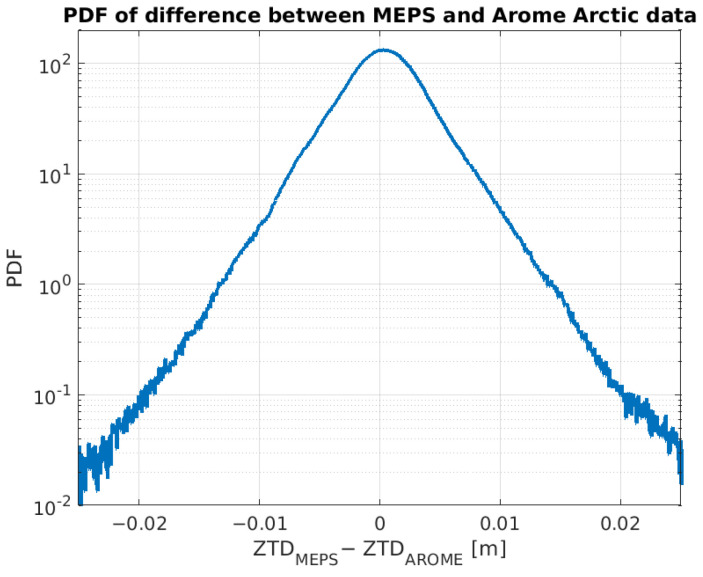
PDF of the difference between calculated ZTD based on MEPS data and on AROME Arctic data.

**Figure 4 sensors-23-01237-f004:**
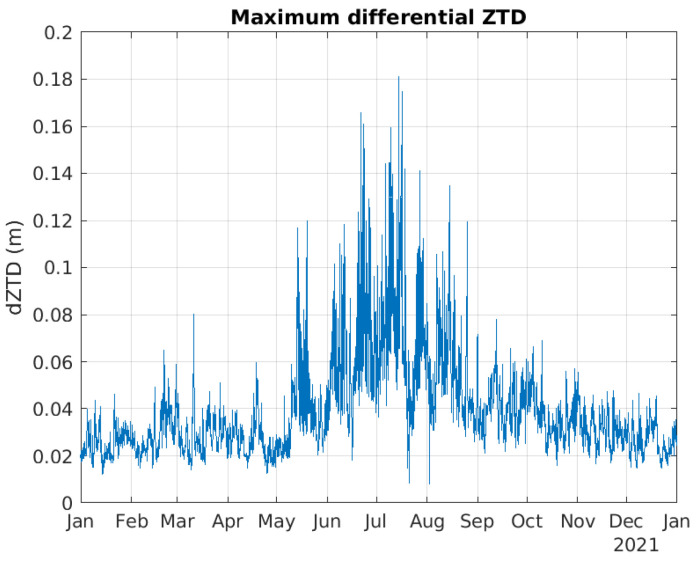
Maximum dZTD values as a function of time.

**Figure 5 sensors-23-01237-f005:**
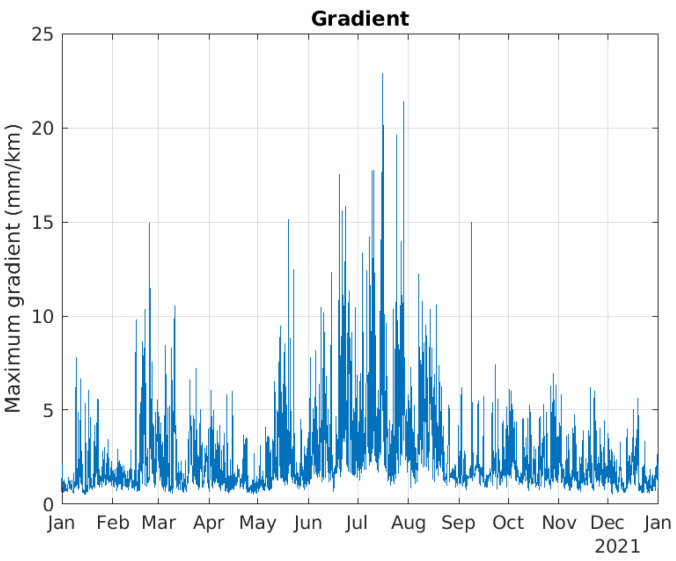
Maximum gradient as a function of time.

**Figure 6 sensors-23-01237-f006:**
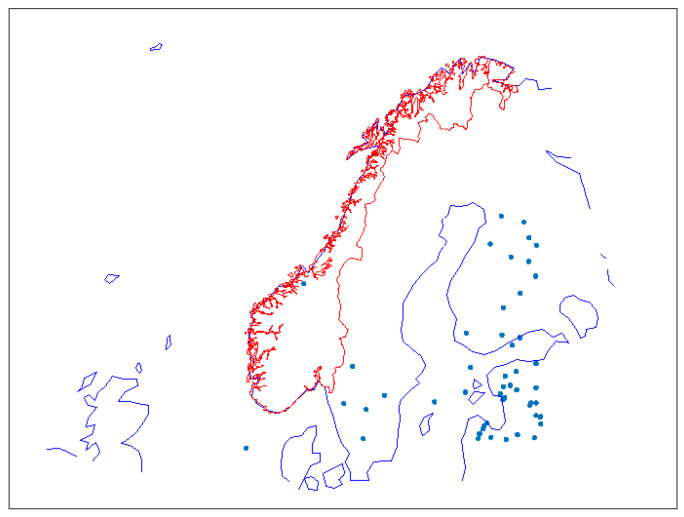
Fifty events with the highest dZTD values.

**Figure 7 sensors-23-01237-f007:**
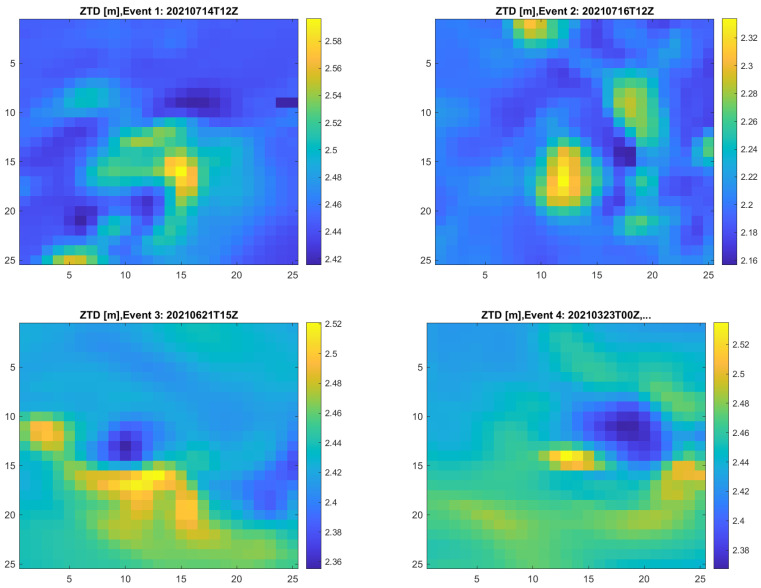
ZTD values for the four largest events.

**Figure 8 sensors-23-01237-f008:**
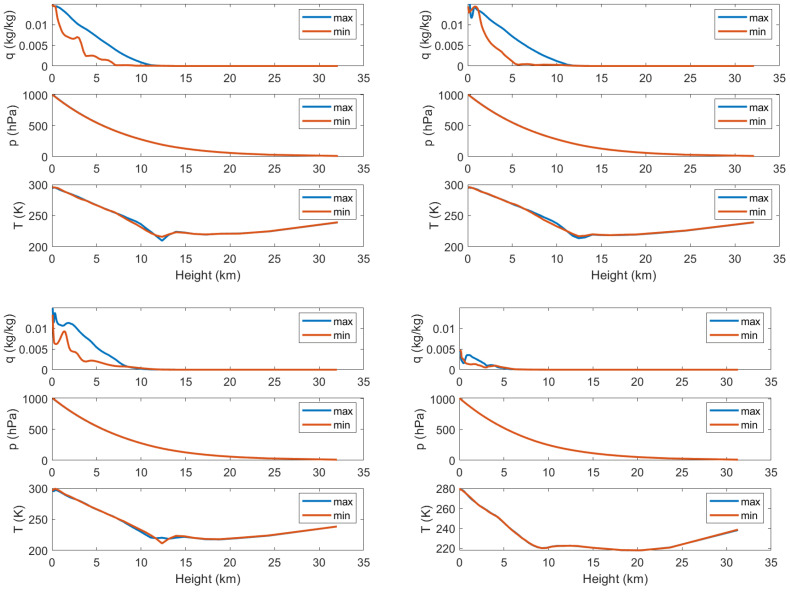
Specific humidity *q*, pressure *p*, and temperature *T* as functions of height for the three pixels with the highest dZTD values, and a pixel with no events (lower right).

**Figure 9 sensors-23-01237-f009:**
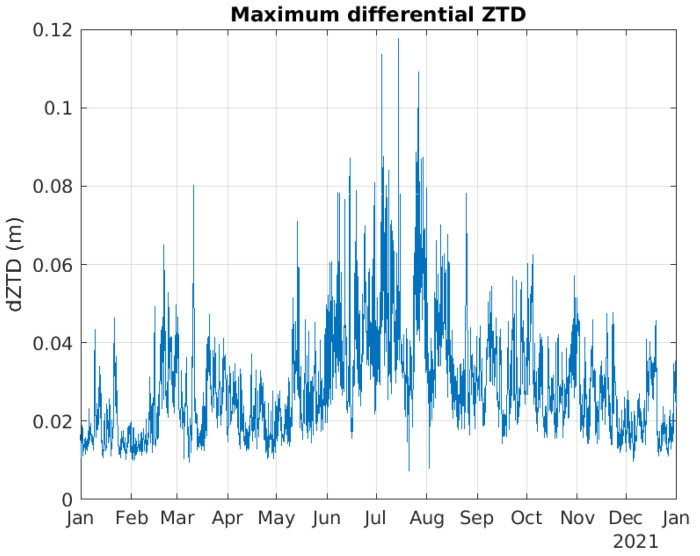
Estimated dZTD values for Norway.

**Figure 10 sensors-23-01237-f010:**
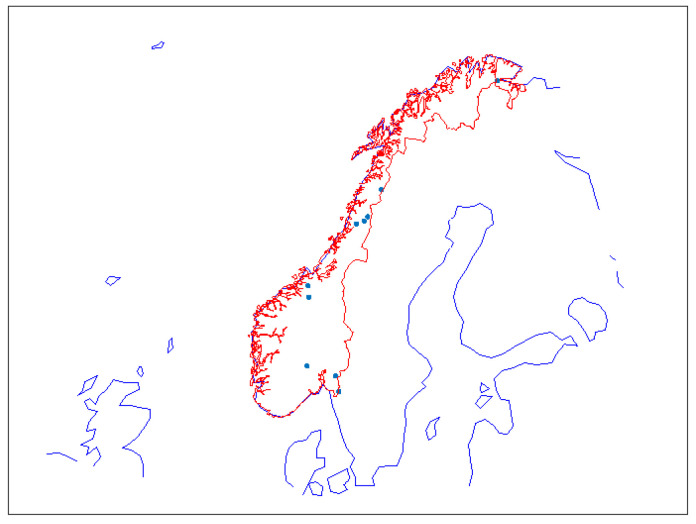
Locations of 10 events with the highest dZTD values in Norway.

**Figure 11 sensors-23-01237-f011:**
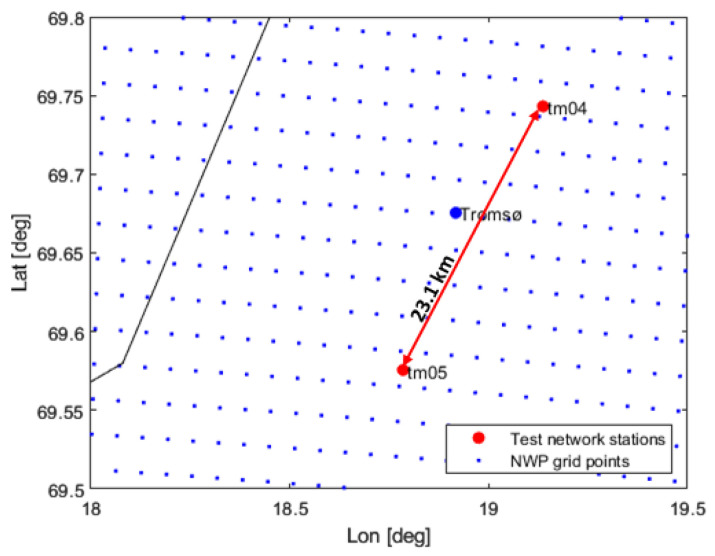
Locations of two GNSS stations and the baseline between them plotted in red; the NWP grid points are shown in blue. The black line is the coastline.

**Figure 12 sensors-23-01237-f012:**
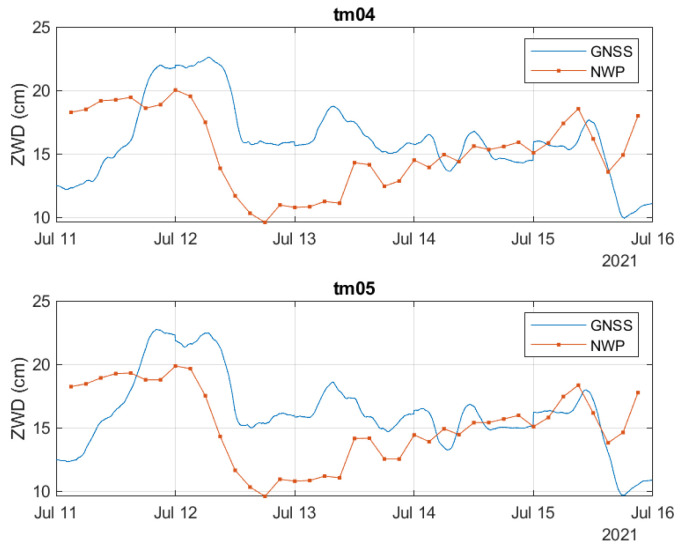
Comparison between GNSS-derived and NWP-model-based measurements.

**Figure 13 sensors-23-01237-f013:**
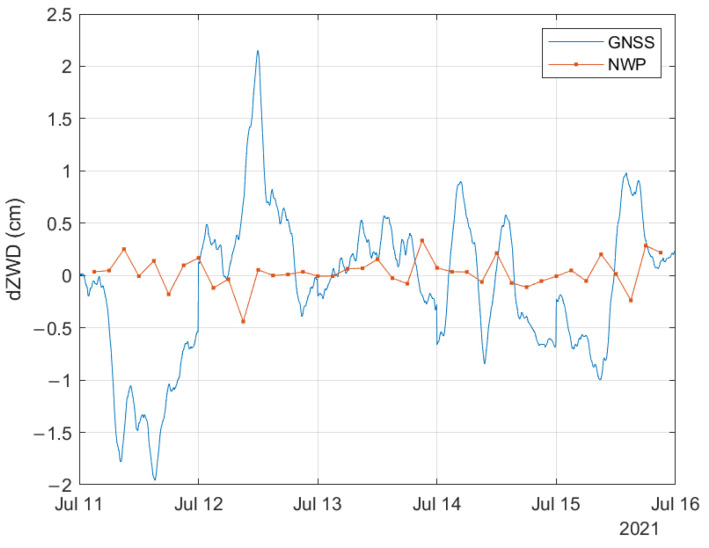
Comparison of dZWD obtained from GNSS derived measurements and NWP model.

**Table 1 sensors-23-01237-t001:** List of meteorological parameters.

Symbol	Name
*t*	time
lat	latitude
lon	longitude
ap	vertical hybrid model
*b*	vertical hybrid model
ps [Pa]	surface air pressure
$\theta_{0}$ [m${}^{2}$/s${}^{2}$]	surface geopotential
*q* [kg/kg]	model level specific humidity
*T* [K]	model level air temperature

**Table 2 sensors-23-01237-t002:** List of dZTD events.

	Time	dZTD (m)	Baseline (km)
1	14-July-2021 12:00:00	0.181	17.7
2	16-July-2021 12:00:00	0.175	14.6
3	21-June-2021 15:00:00	0.166	10.6
4	23-June-2021 00:00:00	0.161	12.5
5	09-July-2021 12:00:00	0.159	10.0

**Table 3 sensors-23-01237-t003:** List of dZTD events in Norway.

	Time	dZTD (m)	Baseline (km)
1	14-July-2021 12:00:00	0.118	13.5
2	04-July-2021 12:00:00	0.114	13.5
3	26-July-2021 18:00:00	0.109	10.3
4	04-July-2021 15:00:00	0.092	20.0
5	26-July-2021 15:00:00	0.091	20.7

**Table 4 sensors-23-01237-t004:** Mean and standard deviation of the difference between ZWD values based on GNSS derived measurements and NWP data.

	tm04	tm05
mean$\left(ZWD_{GNSS}-ZWD_{NWP}\right)$	−1.1 cm	−1.2 cm
std$\left(ZWD_{GNSS}-ZWD_{NWP}\right)$	4.0 cm	3.6 cm

## Data Availability

The NWP data sets used in this study are available at https://thredds.met.no/thredds/metno.html (accessed on 24 November 2022).

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
