# Peer review of "Analysis of Spatial Decorrelation of Small-Scale Tropospheric Delay Using High-Resolution NWP Data†"

_sensors, 2023, doi:10.3390/s23031237_

Round 1
Reviewer 1 Report
Fig. 1: colored squares not very clear P3l86: instead of just saying ‘vast majority of grid points’ please add the empirical probability The validation of the 2 hypotheses in Section 3.1 is not very rigorous. Conclusion that hypothesis 1 is not valid is made based on Fig.7, but you do not explain what is to be seen if it was an edge effect. Can you show that (e.g., with simulation). Also quite interesting from this figure that the effect is very local. Is that something you can really explain meteorologically? Next you show the meteorological parameters (observations?) as function of height for some pixels, but it is not clear how this confirms hypothesis 2. Can also be an error in the observations of q? Suggestion: also make spatial plot of q (like in fig. 7, but then q instead of ZTD). Fig.8: not sharp, fonts too small, lines too thin. Table 4: show standard deviation instead of variance I don’t see how Section 4 contributes. It is a too limited comparison (one baseline, short timeframe) and it seems not related to the events in previous sections. Also the results between GNSS and NWP do not match well, as is clear from the figures and high variance. Regarding the results: how do you take into account height differences between the stations? In conclusion: the manuscript is well-written, methodology is sound, the subject is of interest to the community. However, the presented work is somewhat immature in terms of validation of the results and comparison of GNSS- and NWP-derived ZTDs. Also, authors should more clearly describe the relevance. It is now stated that the phenomena might have large impact on GNSS-based navigation with integrity support. But could this also be dealt with by estimating the dZTD as an extra model parameter as is common nowadays?Author Response
- 1: colored squares not very clear
- Added red and green thick lines. Difficult with AROME Arctic, as the area is not a square in this projection. But hopefully the figure is clearer now.
- P3l86: instead of just saying ‘vast majority of grid points’ please add the empirical probability
- The empirical probability is included
- The validation of the 2 hypotheses in Section 3.1 is not very rigorous.
- See answers to comments 4 and 6 below.
- Conclusion that hypothesis 1 is not valid is made based on Fig.7, but you do not explain what is to be seen if it was an edge effect. Can you show that (e.g., with simulation).
- Our idea was that if the pixels at the edges showed extreme values, the reason might be that the data sources or algorithms used by MET Norway lead to inaccurate results at the edge of coverage. Fig. 7 shows 60x60 km areas containing the largest dZTD values observed over the coverage area. Looking more closely at the yellow and dark blue colours of the figures reveals that the extreme ZTD values are observed at quite a long distance from the edge of coverage. We therefore conclude that the observations most likely are not due to edge effects. This is briefly explained in lines 191-194 in the paper.
- Also quite interesting from this figure that the effect is very local. Is that something you can really explain meteorologically?
- It is not very surprising that events providing the largest dZTD values are local of nature, as more constant ZTD values would give lower dZTD values. Some text is included on this (lines 194-197).
- Next you show the meteorological parameters (observations?) as function of height for some pixels, but it is not clear how this confirms hypothesis 2.
- The ZWD is propositional to the specific humidity q integrated along the vertical column (see eq. 15). Fig. 7 shows how q (which is provided by MET Norway) varies for two locations with large difference in ZWD, while the other key parameters p and T are similar. Hence, Fig. 7 confirms that meteorological effects (that is, the specific humidity) is the cause of the local variations in ZWD.
- Can also be an error in the observations of q?
- All the meteorological parameters provided by MET are estimations. q is one of the parameters provided by MET and may of course be inaccurate. As ground truth data are not available, we don't have any way to determine the accuracy of the parameters.
- Suggestion: also make spatial plot of q (like in fig. 7, but then q instead of ZTD).
- It is not clear how this will provide additional information. ZTD data are accumulated values along the vertical column and hence two dimensional. The q-parameter is three dimensional, with elements over the horizontal grid and at the vertical layers of the model. It will therefore be difficult to visualise the spatial variations of q in a two-dimensional plot. We do however know that the ZWD is proportional to q, and that the other key parameters p and T are relatively constant compared to q (also in areas with large variations in ZTD).
- 8: not sharp, fonts too small, lines too thin.
- Ok, new figures with wider lines and larger font size are included.
- Table 4: show standard deviation instead of variance
- Ok, the table is updated
- I don’t see how Section 4 contributes. It is a too limited comparison (one baseline, short timeframe) and it seems not related to the events in previous sections.
- These are preliminary results. Results based on longer timeframes and with more baselines may be published at a later stage. Still, we think that the comparison gives an indication of the accuracy of using NWP data to estimate tropospheric delay. The GNSS reference station locations are fixed, and it is therefore not surprising that the locations do not coincide with the positions of extreme events.
- The results between GNSS and NWP do not match well, as is clear from the figures and high variance.
- Yes, this is an issue to be looked further into. Both the NWP data and the processed GNSS measurements carry some inaccuracy. How both NWP data with complete and dense coverage and GNSS measurements with limited coverage can be combined to provide the best performance is a topic for further research.
- Regarding the results: how do you take into account height differences between the stations?
- The difference in height between the two stations is about 12 meters (39.29 m vs. 51.46 m). This is not taken into account in the calculation of dZWD. Considering that the ZWD is less than 25 cm for the entire troposphere, it is assumed that the contribution from this height difference is sufficiently small to be neglected for our purposes. In a wider study with more stations with a larger range of heights, this aspect will be included.
- In conclusion: the manuscript is well-written, methodology is sound, the subject is of interest to the community. However, the presented work is somewhat immature in terms of validation of the results and comparison of GNSS- and NWP-derived ZTDs.
- Yes, this part of the paper contains preliminary results, adding to the results presented at the ICL-GNSS conference in June 2022.
- Also, authors should more clearly describe the relevance. It is now stated that the phenomena might have large impact on GNSS-based navigation with integrity support. But could this also be dealt with by estimating the dZTD as an extra model parameter as is common nowadays?
- While it is correct that in some techniques, such as PPP or PPP-RTK, the tropospheric error is being estimated on the user side, this is not the approach taken by Network RTK and similar, where the atmospheric biases are interpolated to the user location. In the case of PPP/PPP-RTK, the vertical tropospheric gradient is the primary concern, however in high dynamic user scenarios the horizontal variation is also likely of interest. In this paper, we focus on characterization of the horizontal dZTD with regard to NRTK and similar system operation, in particular if and when integrity support is considered. Some text is included regarding this (lines 30-36)
Reviewer 2 Report
Review
Zenith total delay (ZTD) is an essential tropospheric parameter and is also very important for precipitable water vapor retrieval. The manuscript investigates the spatial decorrelation of the differential ZTD using the NWP data. The authors investigate the variation of the meteorological parameters during the strongest dZTD events, and preliminarily reveal the correlation between the two datasets. This is an interesting study for analyzing small scale ZTD variations. However, some important clarifications are needed (see below for specific comments). Therefore, I recommend that the manuscript can be accepted after minor revisions.
Specific Remarks
1. Abstract: The abstract is short, and some important experimental results have not been mentioned. Please expand it.
2. Fig.4: Which height does this figure correspond to? Moreover, what is the motivation for calculating dZTD at seven heights? The author never mentioned them again in the later part of the manuscript.
3. Fig.7: What do the X and Y axes of the figures represent, respectively?
4. Equation (18): Did the author propose this formula for the first time? Or someone else has used it. Please provide references. Besides, the rationality of this formula is questionable.
5. Line 245: Indeed, I've been wondering what are the maximum or minimum dZTD values within a 20×20 km pixel. How to determine them? It seems that one pixel only corresponds to one dZTD value.
Author Response
- Abstract:The abstract is short, and some important experimental results have not been mentioned. Please expand it.
- The abstract is expanded, including some results.
- 4: Which height does this figure correspond to? Moreover, what is the motivation for calculating dZTD at seven heights? The author never mentioned them again in the later part of the manuscript.
- For each pixel, the height providing the highest value is selected. This will generally be the lowest height above the surface height.
- There are several reasons why this is done. The main one is that the surface height over the coverage area varies, and the model does correctly not give any result for heights below the surface height. In addition, this allows for comparisons with GNSS-based delays where the GNSS receiver height may vary. The number of heights is limited to seven due to the long processing time.
- 7: What do the X and Y axes of the figures represent, respectively?
- These are maps, each covering 60x60 km. The x-axis roughly corresponds to east-west axis and the y-axis to the south-north axis (depending on the lambert conic projection). The colormap provides the ZTD values for each 2.5x2.5 km pixel.
- Equation (18): Did the author propose this formula for the first time?Or someone else has used it. Please provide references. Besides, the rationality of this formula is questionable.
- This formula is just the first order approximation of a gradient a=Δy/Δx
Round 2
Reviewer 1 Report
thanks for your responses and modifications. Although I still think the results are very preliminary and preferably more experimental results and analysis should be included, I find the current work enough to be accepted. I would recommend to extend your conclusions section with a more specific outlook and what is needed in terms of further work.
Author Response
Thank you for your comments. We have extended the last section and renamed it Conclusions and Future Work. The updates are as follows: The first paragraph is old text with just a couple of minor edits. The second one contains new material describing future work.